# Offset nail fixation for intertrochanteric fractures improves reduction and lag screw position

Takehiro Matsubara[1,2], Kazuhito Soma[3], Ikufumi Yamada[4], Hiroshi Fujita[5], Junya Yoshitani[6], Hiroyuki Oka[7], Hiroyuki Okada[8], Sakae Tanaka[2]*

1 Department of Emergency and Critical Care Medicine, The University of Tokyo Hospital, Bunkyo-ku, Tokyo, Japan, 2 Department of Orthopaedic Surgery, Faculty of Medicine, The University of Tokyo, Bunkyo-ku, Tokyo, Japan, 3 Towa Hospital, Adachi-ku, Tokyo, Japan, 4 Fushimi Momoyama General Hospital, Fushimi-ku, Kyoto, Japan, 5 Hip Joint Center of Rakuyo Hospital, Sakyo-ku, Kyoto, Japan, 6 Takaoka Saiseikai Hospital, Takaoka, Toyama, Japan, 7 Faculty of Medicine, Department of Medical Research and Management for Musculoskeletal Pain, 22nd Century Medical and Research Center, The University of Tokyo, Bunkyo-ku, Tokyo, Japan, 8 Center for Disease Biology and Integrative Medicine, Graduate School of Medicine, The University of Tokyo, Tokyo, Japan

* tanakas-ort@h.u-tokyo.ac.jp

**Data Availability Statement:** The data underlying the results presented in the study are available from Supporting Information.

## Abstract

### Background

Surgery for intertrochanteric fractures using intramedullary hip nails (IHNs) is among the most common surgical procedures in the orthopedic field. Although IHNs provide good overall outcomes, they sometimes cause complications, such as loss of reduction and cut-out. Here, we investigated the usefulness of IHNs with an anterior offset (Best Fit Nail® [BFN]) in maintaining fragment reduction and ensuring proper lag screw position compared with conventional non-offset nails (Proximal Femoral Nail Antirotation® [PFNA]), using postoperative computed tomography (CT).

### Methods

Fifty consecutive patients with intertrochanteric fractures who underwent surgery with BFNs (BFN group) and 50 patients who underwent surgery with PFNAs (PFNA group) were retrospectively analyzed. Indices evaluated by postoperative CT were displacement distance of proximal fragment relative to distal fragment, reduction status (intramedullary, anatomical, and extramedullary types), lag screw direction, and angle between lag screw and femoral neck axis (deviation angle).

### Results

Median [interquartile range] displacement distance was significantly smaller in the BFN group (0 [0, 0] mm) compared with the PFNA group (5.2 [3.6, 7.1] mm) (p<0.001). Reduction status was significantly better in the BFN group (anatomical type, 40 cases; intramedullary type, in 9 cases, and extramedullary type in 1 case) than in the PFNA group (anatomical type, 6 cases; intramedullary type, 43 cases; extramedullary type, 1 case) (p<0.001).

**Funding:** This work was supported by a Grant-in-Aid from the Ministry of Health, Labour and Welfare: 201909030A (Director, Sakae Tanaka).

**Competing interests:** The authors have declared that no competing interests exist.

**Abbreviations:** IHN, intramedullary hip nail; CT, computed tomography; BFN, Best Fit Nail; PFNA, Proximal Femoral Nail Antirotation.

Deviation of lag screw direction was observed in significantly fewer cases in the BFN group (20 cases; 40%) compared with the PFNA group (36 cases; 72%). Lag screw deviation angle was significantly smaller in the BFN group (−0.71˚±4.0˚) compared with the PFNA group (6.9˚±7.1˚). No adverse events related to surgery were observed in either group.

## Conclusions

Intertrochanteric fracture surgery using offset BFNs exhibited significantly smaller displacement distance, better reduction status, and higher frequency of no deviation with central lag screw position, compared with surgery using non-offset PFNAs.

## Introduction

Surgery for intertrochanteric fractures using intramedullary hip nails (IHNs) is among the most common surgical procedures in the orthopedic field [1–3]. Although IHNs provide good overall outcomes, they sometimes cause complications, such as loss of reduction and cut-out, and are associated with higher reoperation rates than sliding hip screws [4–6]. To obtain good clinical results, proper reduction of the fracture fragments and appropriate position of the lag screw are critical. For fragment reduction, it is important to ensure that the proximal fragment is not located posterior to the distal fragment, to prevent the occurrence of excessive telescoping [7–10]. Regarding the lag screw position, the screw are inserted parallel to the femoral neck axis on the lateral radiographic view, and the screw tip should be placed in the central third of the femoral head [11–16]. However, using computed tomography (CT), it was previously reported that it is difficult to achieve both anatomical reduction and proper lag screw position in IHN surgery using conventional non-offset nails [17]: only 18.0% of cases had anatomical reduction and 12.0% had proper lag screw position. It was speculated that the poor reduction and inappropriate lag screw position frequently observed after surgery with IHNs arose through an anterior offset of the femoral neck relative to the femoral shaft axis [17].

To overcome these limitations, we developed an IHN with an anterior offset (Best Fit Nail® [BFN]). The BFN has a lag screw hole with a selectable anterior offset to the nail axis, allowing compensation for deviation between the femoral neck axis and the femoral shaft axis (**Fig 1**). The objective of the present study was to evaluate the usefulness of BFNs in maintaining fragment reduction and ensuring proper lag screw position compared with conventional non-offset nails (Proximal Femoral Nail Antirotation® [PFNA]), using postoperative CT.

## Patients and methods

### Patients and surgical procedures

This retrospective study was conducted at two hospitals in Japan, after receiving approval from the Ethics Review Board at The University of Tokyo (IRB# 2674). Fifty consecutive patients with intertrochanteric fractures who underwent surgery using BFNs (BFN group) from 2019 to 2021, and 50 patients with intertrochanteric fractures who underwent surgery using non-offset PFNAs (PFNA group) from 2010 to 2021 were included. The non-offset PFNA types were determined by the surgeon's preference or the hospital's policy. Patients who required open reduction were excluded from the study. The AO classification was used to evaluate the fracture types.

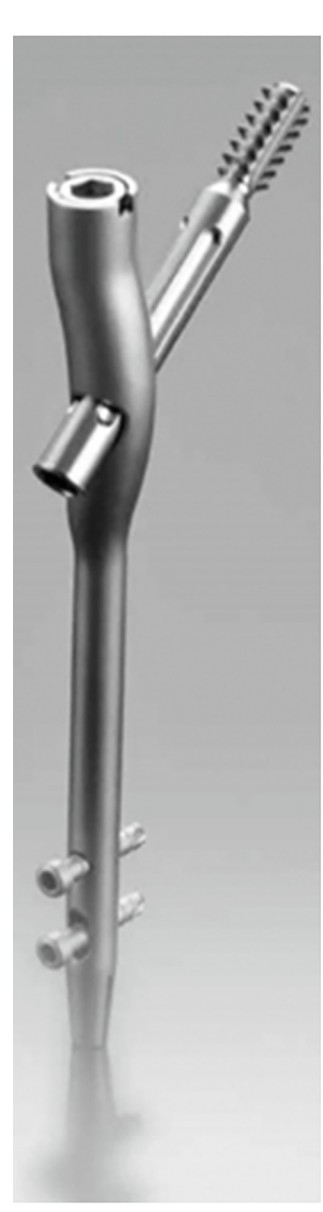

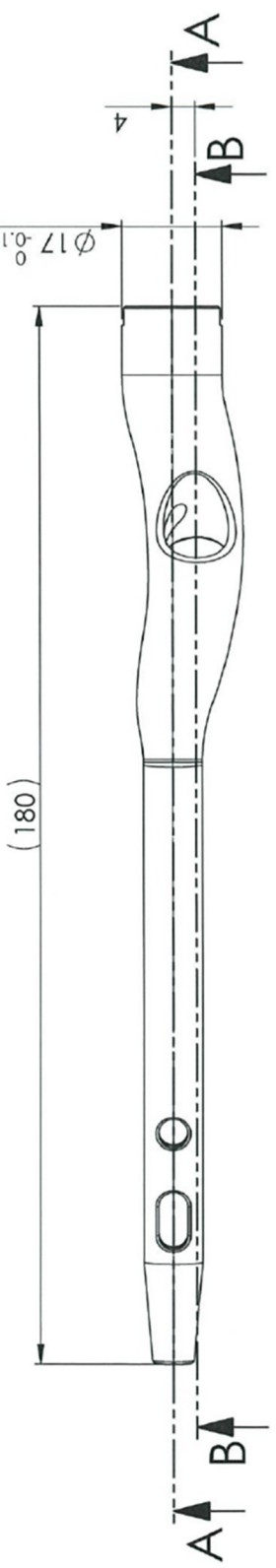

Posterior ⟷ Anterior

**Fig 1. Appearance of the Best Fit Nail.** The Best Fit Nail (BFN) has a lag screw hole with a selectable anterior offset to the nail axis. The anterior offset of the BFNs used in the present study was 4 mm. (A) Nail central axis. (B) Lag screw hole central axis.

All surgeries were performed in the supine position on a traction table, with the aid of an image intensifier system. Closed reduction was performed in a standard manner by traction and adduction of the leg, and the reduction was confirmed by anteroposterior and lateral images [4]. Following insertion of the femoral nail from the tip of the greater trochanter, the lag screw was introduced. Postoperative CT images were obtained using a Canon Alexion TSX (Canon Medical Systems, Tokyo, Japan), with a slice thickness of 1 mm. Nails with 4.0-mm anterior offset were used in the BFN group.

## CT evaluations

The femoral neck was divided into three areas according to the CT axial plane (anterior, central, and posterior). The following indices were evaluated (**Fig 2**): displacement distance, reduction status, and screw direction. Displacement distance was defined as the distance of the proximal fragment relative to the distal fragment on CT (**Fig 2A**). A positive displacement distance indicated posterior displacement of the proximal fragment. Postoperative reduction status was determined using the classification of Ikuta et al. [7,8,10,18,19] (**Fig 2B**). The reduction status was classified as "anatomical type" when there was no displacement of the anterior walls of the fragments, and as "intramedullary type" or "extramedullary type" when the anterior wall of the proximal fragment was displaced posteriorly or anteriorly relative to the anterior wall of the distal fragment, respectively (**Fig 2B**). Screw direction was categorized by the insertion and tip of the lag screw as previously reported [17] (**Fig 2C**). When the insertion and tip of the screw were placed in the same area, the screw direction was considered to have no deviation, and was categorized according to its position (anterior, central, or posterior). The screw direction was considered to show deviation when the insertion and tip were placed in different areas and was categorized as AP (from anterior to posterior) or PA (from posterior to anterior) according to the direction (**Fig 2C**). The angle between the lag screw and the femoral neck axis (deviation angle) was also measured, with a positive value reflecting an anterior direction of the screw.

## Statistical analysis

Demographic data were evaluated using the Shapiro–Wilk test, as well as by observing the kurtosis and skewness of histograms, to determine whether continuous data were normally distributed. The $t$-test was performed for continuous data and the chi-square test was performed for discrete data. Statistical significance was defined as values of $p < 0.05$.

Screw direction was initially classified according to the presence or absence of deviation. Patients with screw deviation were subdivided into two groups: PA deviation and AP deviation. Patients without deviation were subdivided into three groups according to the screw position: anterior, central, and posterior. Therefore, a total of five groups were analyzed. Because the values of the cells in the contingency table were expected to be <10, Fisher's exact test was used to compare the deviation distribution between groups, rather than the chi-square test. Reduction status was categorized into three types: anatomical, intramedullary, and extramedullary, and the distribution was compared between groups using Fisher's exact test. All statistical analyses were conducted using R 4.2.0 (R Core Team 2022) and package tableone 0.13.2.

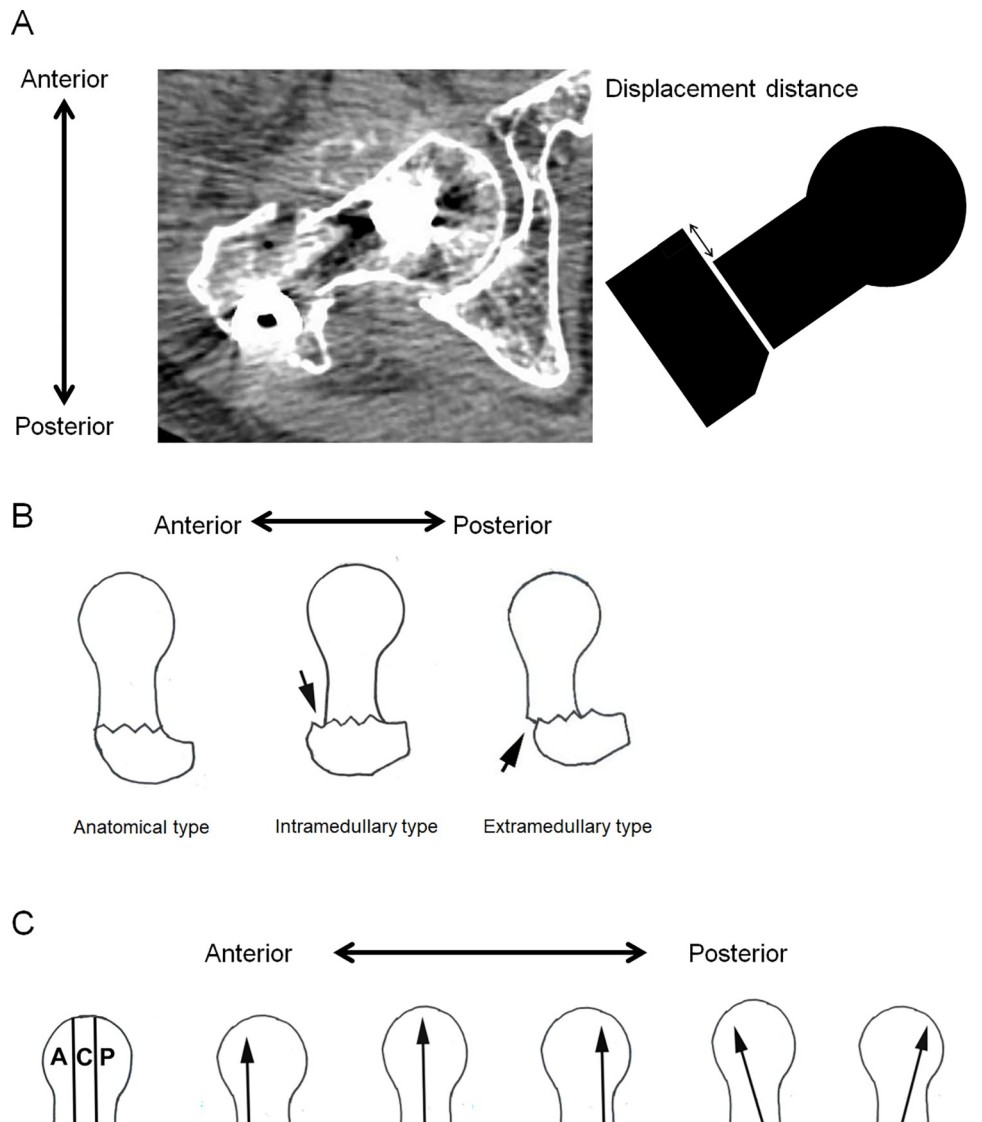

**Fig 2. Computed tomography indices.** (A) Displacement distance. The displacement distance of the proximal fragment relative to the distal fragment was measured at the anterior wall of the fracture site. (B) Reduction status. The reduction status was classified as "anatomical type" when there was no displacement of the anterior walls of the fragments, and as "intramedullary type" or "extramedullary type" when the anterior wall of the proximal fragment was displaced posteriorly or anteriorly relative to the anterior wall of the distal fragment, respectively. (C) Screw direction. The screw direction was categorized according to the insertion and tip of the lag screw. When the insertion and tip of a screw were placed in the same area, the screw direction was considered have no deviation, and was categorized according to its position (A: Anterior; C: Central; P: Posterior). The screw direction was considered to show deviation when the insertion and tip were placed in different areas and was categorized as AP (from anterior to posterior) or PA (from posterior to anterior) according to the direction.

**Table 1. Demographic characteristics of the patients in the BFN and PFNA groups.**

|  |  | BFN group (N = 50) | PFNA group (N = 50) | Pp-value |
|---|---|---|---|---|
| Age |  | 87.0±8.9 | 86.3±7.7 | 0.641 |
| Sex | Female: Male | 39:11 | 39:11 | 1.000 |
| Height |  | 150.7±9.2 | 148.2±10.3 | 0.214 |
| Weight |  | 46.7±11.5 | 47.5±10.6 | 0.714 |
| AO classification | 31-A1 | 7 | 12 | 0.330 |
|  | 31-A2 | 40 | 37 |  |
|  | 31-A3 | 1 | 1 |  |
|  | 31-B2 | 2 | 0 |  |

Continuous and categorical data were analyzed by the *t*-test and Fisher's exact test, respectively.

BFN, Best Fit Nail; PFNA, Proximal Femoral Nail Antirotation.

## Results

### Baseline characteristics

**Table 1** shows the demographic characteristics of the patients in the BFN and PFNA groups. The BFN group comprised 39 women and 11 men with a mean age ± standard deviation of 87.0±8.9 years, while the PFNA group comprised 39 women and 11 men with a mean age ± standard deviation of 86.3±7.7 years. There were no significant differences in age, sex, body height, body weight, or AO classification between the two groups. The neck shaft angle of the nails in the BFN group was 130˚ in all cases. In the PFNA group, the neck shaft angle was 125˚ in all cases.

### CT evaluation

Histograms were created to visualize the displacement distance distribution in the two groups. In the BFN group, the median [interquartile range] displacement distance was 0 [0, 0] mm (**Fig 3A**), and the reduction status was anatomical type in 40 cases, intramedullary type in 9 cases, and extramedullary type in 1 case (**Table 2**). In the PFNA group, the median [interquartile range] displacement distance was 5.2 [3.6, 7.1] mm (**Fig 3B**), and the reduction status was anatomical type in 6 cases, intramedullary type in 43 cases, and extramedullary type in 1 case (**Table 2**). In the BFN group, deviation in the lag screw direction was observed in 20 cases (40%): PA in 6 cases and AP in 14 cases (**Table 2**). In the PFNA group, deviation was observed in 36 cases (72%): PA in 32 cases and AP in 4 cases (**Table 2**). The deviation angle of the lag screw was −0.71˚±4.0˚ in the BFN group and 6.9˚±7.1˚ in the PFNA group. On statistical analysis, the BFN group exhibited significantly less posterior displacement (p<0.001), better reduction status (p<0.001), higher frequency of no deviation with central lag screw position (p<0.001), and smaller deviation angle (p<0.001) than the PFNA group. No clinical or radiographical adverse events related to surgery, such as postoperative infection, venous thromboembolism, and perioperative fractures, were observed in either group during the minimum follow-up period of 3 months.

## Discussion

Simultaneous achievement of anatomical reduction and proper lag screw position is difficult in IHN surgeries using conventional non-offset nails [17]. Only 18.0% of cases achieve anatomical reduction, and only 12.0% show proper lag screw position [17], which may be due to an anterior offset of the femoral neck relative to the femoral shaft. In fact, Anastopoulos *et al.*

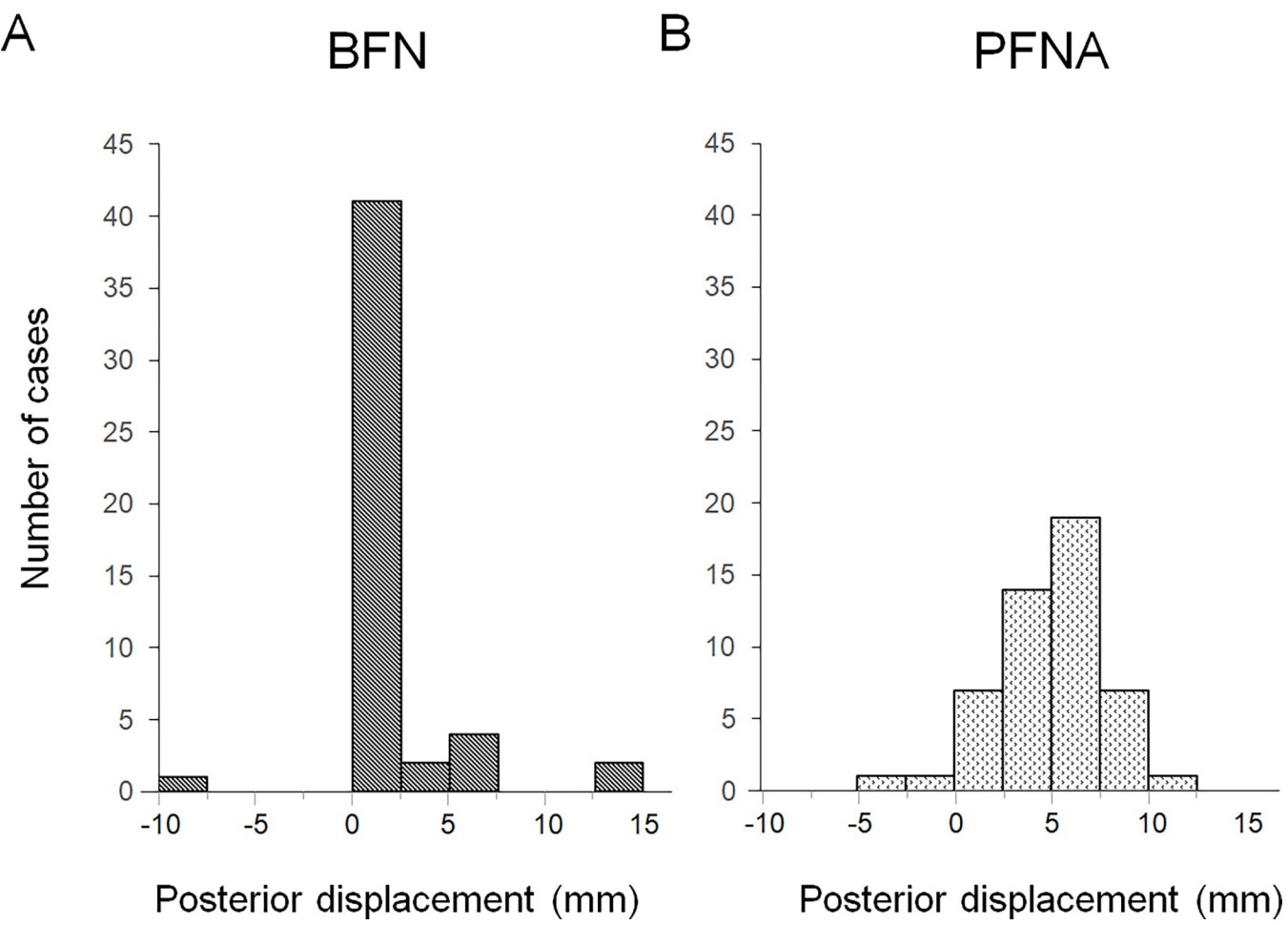

**Fig 3. Distribution of the displacement distance.** (A, B) Histograms showing the displacement distance distributions in the BFN group (A) and the PFNA group (B). Histograms are used in descriptive epidemiological studies. There was a significant difference in the displacement distance distribution between the two groups (p<0.001).

[20] reported that the optimal entry point for IHNs was 3.5±1.5 mm behind the femoral neck axis in the sagittal plane in Greek patients, indicating that the femoral neck was located at an average of 3.5 mm anterior to the femoral shaft axis, reflecting an anterior offset. Similarly, Bonneau *et al.* [21] reported an offset of 4.9±2.6 mm in 50 French and Swiss patients, and Cornelissen *et al.* [22] reported an offset of 6.1±1.7 mm in 100 African patients. In Japanese patients, the anterior offset of the femoral neck relative to the femoral shaft averaged 4.6 mm on CT analysis [17]. Therefore, the femoral neck offsets in various racial groups appear to be comparable.

It has been reported that posterior displacement of the proximal fragment in the lateral radiographic view (intramedullary type) should be avoided in intertrochanteric fracture reduction because it increases the risk of excessive telescoping and cut-out [7,8,10,20]. In surgery with conventional non-offset nail fixation, the lag screw is frequently inserted from the posterior of the femoral neck axis because of the anterior offset of the femoral neck and tends to interfere with the posterior wall of the proximal fragment, thus causing posterior displacement of the fragment (intramedullary type) when the screw is inserted parallel to the femoral neck.

**Table 2. Reduction status and screw direction in the BFN and PFNA groups.**

| BFN group | | | | | | |
|---|---|---|---|---|---|---|
| | | Screw direction | | | | |
| | | No deviation | | | Deviation (+) | |
| | | A | C | P | PA deviation | AP deviation |
| Reduction status | Anatomical type | 0 | 24 | 1 | 3 | 12 |
| | Intramedullary type | 0 | 3 | 1 | 3 | 2 |
| | Extramedullary type | 0 | 1 | 0 | 0 | 0 |
| PFNA group | | | | | | |
| | | Screw direction | | | | |
| | | No deviation | | | Deviation (+) | |
| | | A | C | P | PA deviation | AP deviation |
| Reduction status | Anatomical type | 0 | 1 | 1 | 4 | 0 |
| | Intramedullary type | 0 | 2 | 10 | 27 | 4 |
| | Extramedullary type | 0 | 0 | 0 | 1 | 0 |

The BFN group had higher frequencies of anatomical type as reduction status (p<0.001) and no deviation of the lag screw (p<0.001) compared with the PFNA group.
A: Anterior position; C: Central position; P: Posterior position; AP: Anterior to posterior; PA: Posterior to anterior; BFN: Best Fit Nail; PFNA: Proximal Femoral Nail Antirotation.

To avoid such interference, the lag screw is inserted from posterior to anterior (PA deviation). In fact, PA deviation was observed in 64% of patients in the non-offset PFNA group compared with 12% with the offset BFN group (**Table 2**). However, the force applied by weightbearing may cause posterior displacement of the proximal fragment, resulting in a loss of reduction.

The anatomical offset between the femoral neck axis and the femoral shaft axis is inherently determined. As mentioned above, previous studies reported average anterior femoral neck offsets of 3.5 mm and 4.6 mm [17,20]. Based on these results, we adopted nails with a 4-mm anterior offset in all cases in the present study. We found that the BFN group exhibited significantly better reduction status with a higher anatomical reduction rate (80%) than the PFNA group (12%) (**Table 2**). PA deviation was observed in only 12% of cases in the BFN group. Because the BFN lag screw was inserted close to and parallel to the femoral neck axis on the lateral view, interference between the lag screw and the posterior wall of the femoral neck was rare, resulting in a lower frequency of posterior displacement of the proximal fragment. The median displacement distance in the BFN group was 0 mm, which was significantly better than that in the PFNA group (4.8 mm). The few outliers in the present study may have arisen because a large fragment displacement makes it difficult to achieve reduction with an IHN, and open reduction should have been considered in these cases.

The present study has some limitations. First, the sample size was small. Further studies with larger numbers of patients should be performed to establish the advantages of BFNs over conventional non-offset nails. Second, the size of the anterior offset of the BFNs was fixed at 4 mm. It would be preferable if patient-specific offset sizes could be selected. Third, it remains unclear whether the results obtained in the PFNA group are similar to those for other IHNs. Fourth, we did not analyze the postoperative functional results. Although minimal posterior displacement and straight insertion of the lag screw are considered important, future studies should investigate whether the small posterior displacement and proper lag screw position achieved by BFNs can reduce excessive telescoping and cut-off, resulting in better clinical outcomes. Long-term follow-up of larger numbers of patients is also necessary. Fifth, it is possible that the better reduction in the BFN group was caused by the surgical technique and not by the difference in the nails used. However, all of the surgeries were performed by surgeons

specializing in fracture surgery, and there was a clear difference in the postoperative reduction status between the BFN group and the PFNA group. Thus, we consider that the difference arose from the different designs of the nails. Finally, this was a retrospective observational study conducted in two hospitals, and prospective randomized studies should be performed to demonstrate the superiority of BFNs over non-offset nails in a more precise manner.

Although it is important to avoid unnecessary exposure of patients to radiation, CT examination is very common and reimbursed in Japan for not only orthopaedic departments but also other medical departments, partly due to the high prevalence of CT equipment in hospitals. Moreover, we consider that accurate evaluation of the fracture displacement by CT is very important for postoperative rehabilitation planning. We tried to minimize the radiation exposure to our patients as much as possible, and routine postoperative blood tests revealed that there were no cases of radiation-related complications.

In summary, IHN surgery for intertrochanteric fractures using anterior-offset BFNs exhibited significantly smaller displacement distance, better reduction status, and higher frequency of no deviation with central lag screw position, compared with surgery using conventional non-offset PFNAs. Future prospective studies are required to determine whether surgery using BFNs results in better clinical outcomes than surgery using conventional non-offset nails.

## Conclusions

Intertrochanteric fracture surgery using offset BFNs exhibited significantly smaller displacement distance, better reduction status, and higher frequency of no deviation with central lag screw position, compared with surgery using non-offset PFNAs.

## Supporting information

**S1 Table. Data of BFN group and PFNA group.**
(XLSX)

## Acknowledgments

The authors would like to thank Dr. Kozo Nakamura at Towa Hospital for assistance with the surgeries.

## Author Contributions

**Conceptualization:** Ikufumi Yamada, Sakae Tanaka.

**Data curation:** Hiroyuki Oka, Hiroyuki Okada.

**Funding acquisition:** Sakae Tanaka.

**Investigation:** Takehiro Matsubara, Kazuhito Soma, Ikufumi Yamada, Hiroshi Fujita, Junya Yoshitani.

**Writing – original draft:** Takehiro Matsubara, Sakae Tanaka.

**Writing – review & editing:** Sakae Tanaka.

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
