## [Decision Letter · Decision Letter 0]

1 Sep 2022

PONE-D-22-22038Offset nail fixation for intertrochanteric fractures improves reduction and lag screw positionPLOS ONE

Dear Dr. Tanaka,

Thank you for submitting your manuscript to PLOS ONE. After careful consideration, we feel that it has merit but does not fully meet PLOS ONE’s publication criteria as it currently stands. Therefore, we invite you to submit a revised version of the manuscript that addresses the points raised during the review process.

We look forward to receiving your revised manuscript.

Kind regards,

Faizan Iqbal

Academic Editor

PLOS ONE

Reviewers' comments:

Reviewer's Responses to Questions

**Comments to the Author**

1. Is the manuscript technically sound, and do the data support the conclusions?

Reviewer #1: Partly

Reviewer #2: Partly

Reviewer #3: Yes

2. Has the statistical analysis been performed appropriately and rigorously? 

Reviewer #1: I Don't Know

Reviewer #2: Yes

Reviewer #3: Yes

3. Have the authors made all data underlying the findings in their manuscript fully available?

Reviewer #1: No

Reviewer #2: Yes

Reviewer #3: Yes

4. Is the manuscript presented in an intelligible fashion and written in standard English?

Reviewer #1: Yes

Reviewer #2: No

Reviewer #3: Yes

5. Review Comments to the Author

Reviewer #1: General

1- Is there any published data detected a racial variation in the relation between the neck and shaft of the femur? Are the results obtained from the current study applicable for non– Japanese patients? The authors should discuss this point in the “Discussion”.

2- As the authors of the current manuscript are the developer of the (Best Fit Nail® [BFN]) (Line 74), I wander why they did not plan the study to be a randomized controlled study (RCT) instead of the current retrospective study (with its limitations).

3- Data presented in the manuscript are incomplete and might be deceiving. The results of the manuscript prove that BFN exhibited significantly post-operative smaller displacement distance, better reduction status, and higher frequency of no deviation, with central lag screw positioning, compared to those using non-offset PFNA. What about the differences at the end of follow up? What about the effects of these differences on the clinical results and patient reported outcomes?

Abstract

Adequate.

Line 39-40: “The median displacement distance [interquartile range] was significantly smaller in the BFN group (0 [0, 0] mm) and in the PFNA group (5.2 [3.6, 7.1] mm)” should be corrected to

“The median displacement distance [interquartile range] was significantly smaller in the BFN group (0 [0, 0] mm) than in the PFNA group (5.2 [3.6, 7.1] mm)”

Discussion

Line 185: “To overcome these difficulties, we developed IHNs with a selective anterior…” This is a continuation of the previous paragraph and should not be a new one.

Reviewer #2: Comment on the manuscript Number PONE-D-22-22038

Offset nail fixation for intertrochanteric fractures improves reduction and lag screw Position

• The idea of the research is interesting for the readers of the journal. The authors compared the newly developed nail for fixation of intertrochanteric fracture (which has an anterior offset) so called (Best Fit Nail ® [BFN]) with non-offset Proximal Femoral Nail Antirotation (PFNA). The comparison was based on a post operative CT-scan with focus on maintaining of the reduction of the fracture and the position and the trajectory of the lag screw in the femoral neck. 50 Patients were assigned in each group.

• The authors found that reduction of the fracture and the position of the lag screw were statistically significant better in the BFN group that the TFNA Group.

• Although the Authors tried to describe the methodology in details mentioning the inclusion and exclusion criteria, there are a lot of concerns and questions that have to be clarified:

­ The authors mentioned that the Study was conducted retrospectively. Were the cases consequent? How did they avoid selection bias?

­ The authors conducted CT-Examination for all the patients included in the study. I wonder what was the indication to perform a CT-scan for the patients which is not a routine post operative examination for this type of fractures. This point raises an important ethical consideration: what was the justification to expose the patients to the hazards of CT radiation over the routine postoperative X ray. What consequences did the CT had on the Patient Treatment?

­ It is well known to all orthopedic and trauma surgeon that the reduction of the intertrocateric fractures on the traction table is performed before starting the implantation of the nail. The nail fixes only the achieved reduction. How could the autrhors Proof that the better reduction is related to the nail it self (not the surgical technique) specially when the 2 nails are performed in 2 different clinics by different surgeons.

­ The authors mentioned that there were no adverse effect related to the performed surgeries. this is a broad word mentioned in the results without mentioning how did they look for this in the methodology. Which type of adverse effects were Evaluated (Clinical/ Radiological) ? Intraoperative or Post operative? How was the evaluation of the adverse effects was conducted? how long was the follow up to mention this fining in the outcome?

• The article is written in a clear and simple English language. Yet there are some spelling, grammatical and punctuation mistakes:

­ Line 34: Antirotatin should be antirotation

­ Line 152: one case not 1 cases.

­ I recommend the article and the punctuation to be revised by an english language specialist.

• The included Diagrams and table are informative.

• The abstract is short and comprehensive and summarizes the important points of the article.

• The authors reported the strength and weak points and the limitation of the study.

•

• Decision:

According to the guidelines of the critical appraisal, i recommend the authors to resubmit the article after considering the previously mentions major improvement recommendations.

Reviewer #3: I think this new design should be used in a wider scale of patients before popularization of the design and technique, however many commercial issues will affect the progress of popularization of the implants, good luck

6. PLOS authors have the option to publish the peer review history of their article (what does this mean?). If published, this will include your full peer review and any attached files.

Reviewer #1: **Yes: **Mohamed Abdel-wanis

Reviewer #2: **Yes: **Dr. Ayman F. AbdelKawi, MD

Reviewer #3: **Yes: **Ahmed H. K. Abdelaal

---

## [Author Response · Author response to Decision Letter 0]

10 Sep 2022

Response to Editor and Reviewers

PONE-D-22-22038

Offset nail fixation for intertrochanteric fractures improves reduction and lag screw position

Dear Editor and Reviewers

We thank the Editor and Reviewers for thoughtful evaluation of our manuscript. We have given careful attention to the suggestions and addressed their comments. Point-by-point responses to specific comments are provided below. The editor’s and reviewers’ comments are in bold, italic and underlined. The changes made in the manuscript were highlighted.

To Editor

PLOS only allows data to be available upon request if there are legal or ethical restrictions on sharing data publicly.

→ We uploaded our raw data as a Supporting Information file, accordingly.

To Reviewer #1: 

General

1) Is there any published data detected a racial variation in the relation between the neck and shaft of the femur? Are the results obtained from the current study applicable for non– Japanese patients? The authors should discuss this point in the “Discussion”.

→ Thank you for the valuable comments. In fact, only a few studies investigated the offset of the femoral neck or its racial differences. Anastopoulos reported the 3.5 ± 1.5 mm offset in Greek patients (Reference #20). Bonneau et al. reported the 4.9 ± 2.6 mm offset in 50 French and Swiss cases (Reference #21). Cornelissen reported the 6.1 ± 1.7 mm offset in 100 African cases (Reference #22). These studies show that femoral neck offsets are comparable among various racial groups. We discussed this point in the Discussion section (page 10 of the revised manuscript).

2) As the authors of the current manuscript are the developer of the (Best Fit Nail® [BFN]) (Line 74), I wander why they did not plan the study to be a randomized controlled study (RCT) instead of the current retrospective study (with its limitations).

→ Thank you for your comment. In this paper, we tried to evaluate the usefulness of BFNs in maintaining the reduction of fragments and ensuring proper lag screw positioning, compared to that with conventional non-offset nails (PFNA) retrospectively. We agree with the reviewer that it is important to confirm the results in this study in an RCT setting in the future.

3) Data presented in the manuscript are incomplete and might be deceiving. The results of the manuscript prove that BFN exhibited significantly post-operative smaller displacement distance, better reduction status, and higher frequency of no deviation, with central lag screw positioning, compared to those using non-offset PFNA. What about the differences at the end of follow up? What about the effects of these differences on the clinical results and patient reported outcomes?

→ Thank you for the valuable comments. As the reviewer pointed, it is important to elucidate the clinical significance between BFN and non-off set nails. So far, we have only preliminary results that patients treated with BFN return to daily activities somewhat faster, which needs to be confirmed with more cases in the future.

4) Line 39-40: “The median displacement distance [interquartile range] was significantly smaller in the BFN group (0 [0, 0] mm) and in the PFNA group (5.2 [3.6, 7.1] mm)” should be corrected to “The median displacement distance [interquartile range] was significantly smaller in the BFN group (0 [0, 0] mm) than in the PFNA group (5.2 [3.6, 7.1] mm)”

→ Corrected, accordingly.

5) Line 185: “To overcome these difficulties, we developed IHNs with a selective anterior…” This is a continuation of the previous paragraph and should not be a new one.

→ We removed the phrase, accordingly.

To Reviewer #2

1) The authors mentioned that the Study was conducted retrospectively. Were the cases consequent? How did they avoid selection bias?

→ Consecutive patients who underwent surgery for intertrochanteric fracture using BFNs or using non-offset PFNA nails were included in this study (page 5), and we did not exclude the bias in advance. However, when the background data were compared, there were no significant differences in age, sex, height, weight, or AO classification between the BFN and PNFA groups as a result, which may be due to the relatively consistent indications for surgery.

2) The authors conducted CT-Examination for all the patients included in the study. I wonder what was the indication to perform a CT-scan for the patients which is not a routine post operative examination for this type of fractures. This point raises an important ethical consideration: what was the justification to expose the patients to the hazards of CT radiation over the routine postoperative X ray. What consequences did the CT had on the Patient Treatment?

→ Thank you for raising an important issue. As the reviewer mentioned, it is important to avoid radiation exposure to the patients. However, CT examination is very common and reimbursed in Japan not only in orthopedic practice but also other medical departments partly because of the high prevalence of CT equipment. In addition, we consider accurate evaluation of the displacement of fractures by CT is very important for planning postoperative rehabilitation. We are trying to minimize the radiation exposure to the patients as much as possible, and routine post-operative blood tests revealed there were no cases of radiation-related complications. 

3) It is well known to all orthopedic and trauma surgeon that the reduction of the intertrochanteric fractures on the traction table is performed before starting the implantation of the nail. The nail fixes only the achieved reduction. How could the authors Proof that the better reduction is related to the nail itself (not the surgical technique) specially when the 2 nails are performed in 2 different clinics by different surgeons.

→ As the reviewer pointed out, it is possible that the better reduction was caused by the surgical technique and not by the difference of nails. However, all the surgeries were performed by surgeons specializing in fracture surgery, and there was a clear difference in the post-operative reduction status between BFN group and PFNA group. Therefore, we consider that this difference is due to the different design of the nails.

4) The authors mentioned that there were no adverse effect related to the performed surgeries. this is a broad word mentioned in the results without mentioning how did they look for this in the methodology. Which type of adverse effects were Evaluated (Clinical/ Radiological) ? Intraoperative or Post operative? How was the evaluation of the adverse effects was conducted? how long was the follow up to mention this fining in the outcome?

→ Thank you for your comments. Clinical (and radiological) adverse events reported by surgeons, such as postoperative infection, venous thromboembolism, and perioperative fractures were counted as adverse events in this study. This is explained in the revised manuscript (page 9 of the revised manuscript).

5) The article is written in a clear and simple English language. Yet there are some spelling, grammatical and punctuation mistakes:

­ Line 34: Antirotatin should be antirotation

­ Line 152: one case not 1 cases.

­ I recommend the article and the punctuation to be revised by an english language specialist.

→ We corrected the typos, accordingly. The article was edited by an English-speaking specialist.

To Reviewer #3: 

I think this new design should be used in a wider scale of patients before popularization of the design and technique, however many commercial issues will affect the progress of popularization of the implants, good luck.

→Thank you for the comments. We also recognize that many commercial issues will affect the progress of popularization of the implants as the reviewer mentioned.

Other changes

1) We made a mistake in the number of patients with 130 and 125 degrees of BFN nails, and corrected (page 9 and Table 1).

2) The mean age of patients in the PFNA group and the number of patients in each of the AO classifications in the BFN group were incorrect and were corrected (Table 1). The data were statistically re-analyzed and the p-values remained unchanged. 

We are sorry for the mistakes.

---

## [Decision Letter · Decision Letter 1]

27 Sep 2022

PONE-D-22-22038R1Offset nail fixation for intertrochanteric fractures improves reduction and lag screw positionPLOS ONE

Dear Dr. Tanaka,

Thank you for submitting your manuscript to PLOS ONE. After careful consideration, we feel that it has merit but does not fully meet PLOS ONE’s publication criteria as it currently stands. Therefore, we invite you to submit a revised version of the manuscript that addresses the points raised during the review process.

We look forward to receiving your revised manuscript.

Kind regards,

Faizan Iqbal

Academic Editor

PLOS ONE

Additional Editor Comments:

Dear author,

I personally go through your manuscript. The topic is interesting but the manuscript is not written in intelligible fashion and I will suggest you to please re-write your manuscript in scientific language. English language can be further polished.

I hope you will address these comments in your future submission.

Thank you

Reviewers' comments:

Reviewer's Responses to Questions

**Comments to the Author**

1. If the authors have adequately addressed your comments raised in a previous round of review and you feel that this manuscript is now acceptable for publication, you may indicate that here to bypass the “Comments to the Author” section, enter your conflict of interest statement in the “Confidential to Editor” section, and submit your "Accept" recommendation.

Reviewer #1: All comments have been addressed

Reviewer #2: (No Response)

Reviewer #3: (No Response)

2. Is the manuscript technically sound, and do the data support the conclusions?

Reviewer #1: (No Response)

Reviewer #2: Partly

Reviewer #3: Yes

3. Has the statistical analysis been performed appropriately and rigorously? 

Reviewer #1: (No Response)

Reviewer #2: Yes

Reviewer #3: Yes

4. Have the authors made all data underlying the findings in their manuscript fully available?

Reviewer #1: (No Response)

Reviewer #2: Yes

Reviewer #3: Yes

5. Is the manuscript presented in an intelligible fashion and written in standard English?

Reviewer #1: (No Response)

Reviewer #2: No

Reviewer #3: Yes

6. Review Comments to the Author

Reviewer #1: (No Response)

Reviewer #2: I thank the authors for their reply, explaining their points of view and their effort to revise the manuscript. Unfortunately, there is still improvement potentials. The Authors answered the questions and commented of the reviewers but did not add their ideas in the manuscript (with few exceptions). For example they did not add follow up Period, instead reported some few possible complication which was excluded clinically by the surgeons in the results section. They did not add the weak points mentioned by the authors in the discussion section.

they commented on the hazards of raditation in their reply but did mention that in they manuscript.

I recommend the authors to add their replay on the reviewers in the discussion section.

additionally i recommend tthe authors to omit the clinical follow up completely and imphasis in the aim of the study on that they were looking only on the CT finidng postoperatively. this may make the article more plausible.

Reviewer #3: (No Response)

7. PLOS authors have the option to publish the peer review history of their article (what does this mean?). If published, this will include your full peer review and any attached files.

Reviewer #1: **Yes: **Mohamed Abdel-wanis

Reviewer #2: **Yes: **Dr. Ayman F. AbdelKawi, MD

Reviewer #3: **Yes: **Ahmed H.K. Abdelaal

---

## [Author Response · Author response to Decision Letter 1]

4 Oct 2022

We thank the Editor and Reviewers for thoughtful evaluation of our manuscript. According to the Editor and Reviewers’ comments, we revised the manuscript. Point-by-point responses to specific comments are provided below. 

To Editor

I personally go through your manuscript. The topic is interesting but the manuscript is not written in intelligible fashion and I will suggest you to please re-write your manuscript in scientific language. English language can be further polished.

→According to the comment, the revised manuscript was edited by an English-speaking scientist. I have not highlighted the text if it is only a wording or grammatical modification.

To Reviewer 2

1) The Authors answered the questions and commented of the reviewers but did not add their ideas in the manuscript (with few exceptions). For example, they did not add follow up Period, instead reported some few possible complication which was excluded clinically by the surgeons in the results section. They did not add the weak points mentioned by the authors in the discussion section. They commented on the hazards of radiation in their reply but did mention that in their manuscript. I recommend the authors to add their replay on the reviewers in the discussion section.

→I apologize that I did not revise the manuscript satisfactorily. According to the reviewer’s comments, we added the following sentences in Results and Discussion sections. 

“No clinical or radiographical adverse events related to surgery, such as postoperative infection, venous thromboembolism, and perioperative fractures, were observed in either group during the minimum follow-up period of 3 months.” (page 11)

“Fifth, it is possible that the better reduction in the BFN group was caused by the surgical technique and not by the difference in the nails used. However, all of the surgeries were performed by surgeons specializing in fracture surgery, and there was a clear difference in the postoperative reduction status between the BFN group and the PFNA group. Thus, we consider that the difference arose from the different designs of the nails.” (page 15-16)

“Although it is important to avoid unnecessary exposure of patients to radiation, CT examination is very common and reimbursed in Japan for not only orthopaedic departments but also other medical departments, partly due to the high prevalence of CT equipment in hospitals. Moreover, we consider that accurate evaluation of the fracture displacement by CT is very important for postoperative rehabilitation planning. We tried to minimize the radiation exposure to our patients as much as possible, and routine postoperative blood tests revealed that there were no cases of radiation-related complications.” (page 16)

2) Additionally I recommend the authors to omit the clinical follow up completely and emphasis in the aim of the study on that they were looking only on the CT finding postoperatively. this may make the article more plausible.

→According to the reviewer’s comments, we removed the following sentences “We have preliminary results suggesting that the reduction of fragments and lag screw positioning were similar between PFNA and other non-offset nail cases, which should be further analyzed. (page 15)”

---

## [Decision Letter · Decision Letter 2]

17 Oct 2022

Offset nail fixation for intertrochanteric fractures improves reduction and lag screw position

PONE-D-22-22038R2

Dear Dr. Tanaka,

We’re pleased to inform you that your manuscript has been judged scientifically suitable for publication and will be formally accepted for publication once it meets all outstanding technical requirements.

Kind regards,

Faizan Iqbal

Academic Editor

PLOS ONE

Additional Editor Comments (optional):

Reviewers' comments:

Reviewer's Responses to Questions

**Comments to the Author**

1. If the authors have adequately addressed your comments raised in a previous round of review and you feel that this manuscript is now acceptable for publication, you may indicate that here to bypass the “Comments to the Author” section, enter your conflict of interest statement in the “Confidential to Editor” section, and submit your "Accept" recommendation.

Reviewer #2: All comments have been addressed

2. Is the manuscript technically sound, and do the data support the conclusions?

Reviewer #2: Yes

3. Has the statistical analysis been performed appropriately and rigorously? 

Reviewer #2: Yes

4. Have the authors made all data underlying the findings in their manuscript fully available?

Reviewer #2: Yes

5. Is the manuscript presented in an intelligible fashion and written in standard English?

Reviewer #2: Yes

6. Review Comments to the Author

Reviewer #2: (No Response)

7. PLOS authors have the option to publish the peer review history of their article (what does this mean?). If published, this will include your full peer review and any attached files.

Reviewer #2: **Yes: **Dr. Ayman F. AbdelKawi, MD.

---

## [Editor Report · Acceptance letter]

20 Oct 2022

PONE-D-22-22038R2 

Offset nail fixation for intertrochanteric fractures improves reduction and lag screw position 

Dear Dr. Tanaka:

I'm pleased to inform you that your manuscript has been deemed suitable for publication in PLOS ONE. Congratulations! Your manuscript is now with our production department. 

Kind regards, 

on behalf of

Dr. Faizan Iqbal 

Academic Editor

PLOS ONE